# Brief self-guided digital intervention versus a comprehensive therapist-guided online cognitive behavioural therapy for atopic dermatitis: a trial protocol for a randomised non-inferiority trial

Dorian Kern ![ORCID] ,[1,2] Brjánn Ljótsson,[2] Louise Lönndahl,[3] Erik Hedman-Lagerlöf,[2] Maria Bradley,[3] Nils Lindefors,[1] Martin Kraepelien[1,2]

¹Centre for Psychiatry Research, Department of Clinical Neuroscience, Stockholm Health Care Services & Karolinska Institutet, Stockholm, Sweden
²Division of Psychology, Department of Clinical Neuroscience, Karolinska Institutet, Stockholm, Sweden
³Dermatology and Venereology Unit, Department of Medicine Solna, Karolinska Institutet, Stockholm, Sweden

**Correspondence to**
Mr Dorian Kern;
dorian.kern@ki.se

## ABSTRACT

**Introduction** Our aim is to investigate whether a shortened digital self-care intervention is non-inferior to, and cost-effective compared with, a comprehensive and therapist-guided cognitive behavioural therapy treatment for atopic dermatitis (AD).

**Methods and analysis** This is a single-blind, randomised clinical non-inferiority trial at Karolinska Institutet, a medical university in Stockholm, Sweden. We will recruit 174 adult participants with AD through self-referral. Participants will be randomised 1:1 to the two experimental conditions. Participants randomised to guided care will receive internet-delivered cognitive behavioural therapy for 12 weeks. Participants randomised to digital self-care will have access to this self-guided intervention for 12 weeks. At post-treatment (primary endpoint), non-inferiority will be tested and resource use will be compared between the two treatment groups. Cost-effectiveness will be explored at 1-year follow-up. Potential mediators will be investigated. Data will be analysed intention to treat. We define non-inferiority as a three-point difference on the primary outcome measure (Patient-oriented Eczema Measure). Recruitment started in November 2022.

**Ethics and dissemination** This study is approved by the Swedish ethics authority (reg. no 2021-06704-01) and is preregistered at ClinicalTrials.gov. The study will be reported according to the Consolidated Standards of Reporting Trials statement for non-pharmacological trials. The results of the study will be published in peer-reviewed scientific journals and disseminated to patient organisations and media.

**Trial registration number** NCT05517850.

## STRENGTHS AND LIMITATIONS OF THIS STUDY

⇒ Includes a full health economics evaluation.
⇒ Broad inclusion criteria will help generalise the results to a variety of atopic dermatitis cases.
⇒ Depending on what time of the year the majority of participants eventually will be recruited, it may be difficult to control for seasonal changes in Sweden's temperate climate.
⇒ Earlier similar studies indicate a risk of bias for the recruitment of female participants with an advanced education.
⇒ If the self-guided condition cannot be found non-inferior, this study cannot answer what specific differences between the conditions caused this result.

## BACKGROUND
### Atopic dermatitis

Atopic dermatitis (AD) is the most prevalent of the inflammatory skin diseases. The primary symptoms of AD are itch, dry skin and rash.[1] Treatment and prevention generally consist of emollient creams, topical corticosteroids, other local immunosuppressive creams such as calcineurin inhibitors. If topical treatment is ineffective, phototherapy and systemic immunomodulatory treatments are often given.[2] In addition to dermatological symptoms, people with AD have an increased risk of major depression and anxiety disorders.[3 4] Furthermore, behavioural and emotional factors may affect AD symptoms. A common reaction to the itching sensation of the skin is the act of scratching, which, while temporarily relieving, can lead to damage to the skin barrier and an aggravated inflammation.[5] Furthermore, people with AD tend to scratch when they experience anxiety or depressed mood,[6] possibly as a conditioned behaviour. In addition, avoidance of itch-associated situations is frequently documented, which may lead to avoidance of activities that are otherwise appreciated or important to the patient, which can further affect mental health and quality of life[7] The

impairment of quality of life by AD is comparable to asthma and type 1 diabetes.[2] The facts that AD can affect mental health, and that behaviour can directly affect AD symptoms, make psychological treatment relevant.

## Educational and psychological interventions for AD

Various forms of educational and psychological support may be available for people with AD. According to structured reviews, studies of educational interventions for AD have often had suboptimal methodology and unclear results, with tendencies towards effect on skin-related quality of life and improved symptoms.[8] These interventions may be held in a group or individual setting and may be led by physicians, nurses or other professionals. One recent randomised controlled trial (RCT) of an internet-delivered educational and behavioural intervention, including young adults, showed small but significant effects on AD symptoms and secondary measures.[9]

## Internet-delivered cognitive behavioural therapy

Internet cognitive behavioural therapy (ICBT) is a psychological treatment administered over the internet, otherwise similar to regular face-to-face CBT. ICBT usually includes therapist support via a secure digital communication, similar to email. Studies have shown that a clinician spends the equivalent of about 15 minutes per patient and week in ICBT.[10] ICBT has been suggested to improve cost-effectiveness as well as the availability of evidence-based treatment[11] and research reviews show that there are no consistent differences in effectiveness between conventional face-to-face CBT and ICBT.[12] In Sweden, ICBT has been offered in regular care via the Internet Psychiatry Unit in Stockholm since 2007.[13]

Although ICBT has many advantages, there is an unmet need for interventions for people with mild to moderate illness, where a comprehensive and therapist-guided ICBT programme would be an unnecessarily intense level of care, and thus an inefficient use of resources. One solution is to use existing ICBT programmes and simply remove the therapist guidance. Some research has suggested that this has a negative impact on effectiveness, however, there are indications that the amount and type of therapist contact are not decisive[14] and recent studies have compared therapist supported ICBT and self-guided ICBT and shown that treatment outcome, treatment implementation and treatment satisfaction do not differ between conditions if certain components are present in the self-guided intervention. These components include a stringent clinical process in the form of an assessment and a follow-up interview with a clinician and a high-quality interface.[15 16]

## Internet-delivered cognitive behavioural therapy for AD

An ICBT protocol has been evaluated for AD. The main components were symptom diary, mindfulness and exposure to avoided sensations and situations, with prevention of unhelpful responses, such as scratching. The treatment showed promising preliminary effects in a face-to-face pilot study, which subsequently was confirmed in a RCT.[17 18] A digital self-care intervention has been preliminarily evaluated for people with AD, with promising effects.[19] Digital self-care was based on the ICBT intervention with comparable components. Thus, there is support that CBT can be effective in improving mental health and reducing symptoms of AD, although additional studies are needed. A comparison between the ICBT and reworked digital self-care variants could further contribute to the literature and the results will have implications for implementation.

This is a study protocol of a single-blind non-inferiority RCT, aimed at evaluating the effects of a self-guided CBT programme available via the internet (digital self-care) for people with AD, in relation to a previously evaluated therapist-guided internet-delivered CBT programme (guided care).[18] The main hypothesis is that digital self-care will be non-inferior to guided care in reducing self-rated eczemic symptoms, as measured by the patient-oriented eczema measure (POEM).[20] Secondary purposes are to evaluate use of resources and cost-effectiveness of digital self-care compared with guided care, and to investigate potential mediators of symptom improvement.

### Hypotheses
#### Primary hypothesis
Digital self-care will be non-inferior to ICBT, according to the predefined non-inferiority margin of three points, as measured by the primary outcome (POEM).

#### Secondary hypothesis 1
Digital self-care will be associated with significantly less staff costs than ICBT, in the form of time spent in hours by a licensed psychologist.

#### Secondary hypothesis 2
Digital self-care will be cost-effective compared with ICBT, assessed at 1-year follow-up.

#### Secondary hypothesis 3
Effects on the outcome will be mediated by the amount of time spent by participants on main treatment components.

### Study design
#### Time frame
The first participant was included on 29 November 2022. The first cohort started on 5 December 2022. Post-treatment data collection is expected to finish summer or autumn 2023, with the last follow-up a year after that. The present study is a non-inferiority RCT for adults with AD. Participants randomised to guided care will receive ICBT via the internet, guided by a therapist for 12 weeks. Participants randomised to digital self-care get access to a self-guided CBT-based intervention for 12 weeks. Condition allocation will be 50/50.

The primary end point is at 12 weeks, (ie, post-treatment). The study will also include a 6-month and a 1-year follow-up. One year after treatment conclusion, long-term effects will be evaluated. An overview of the trial design can be seen in figure 1.

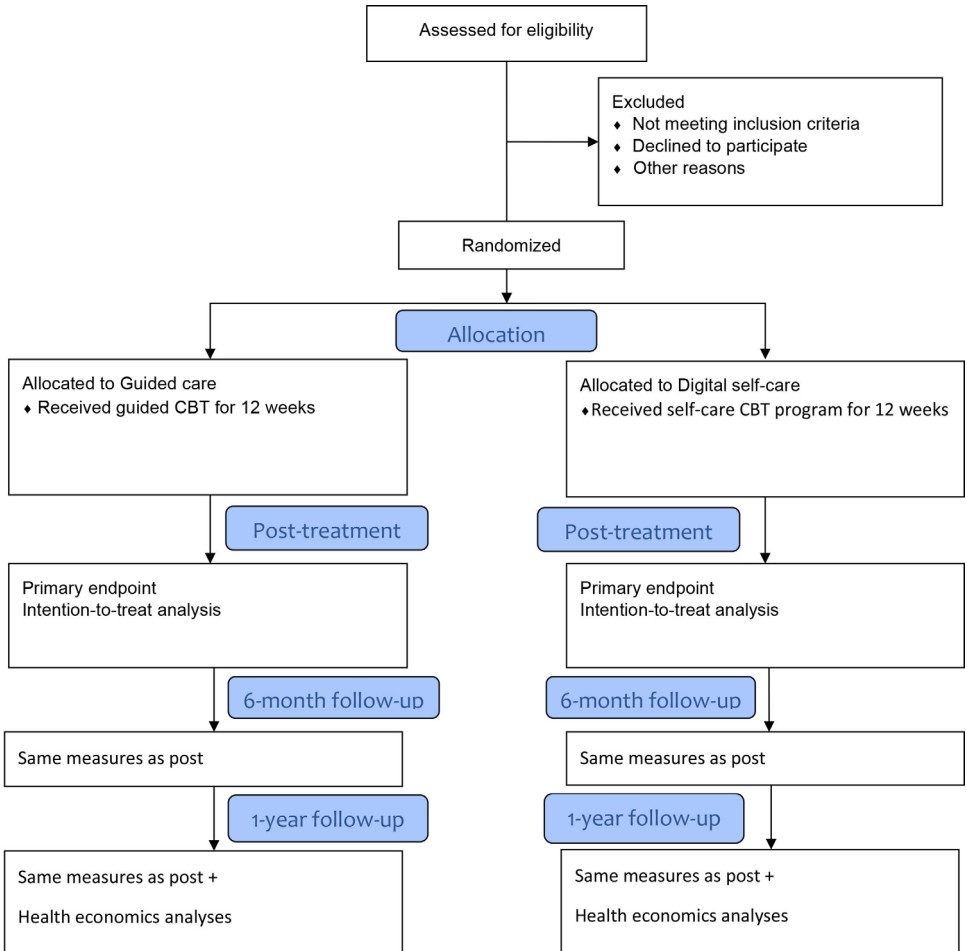

**Figure 1** CONSORT 2010 flow diagram. CBT, cognitive behavioural therapy; CONSORT, Consolidated Standards of Reporting Trials.

The study is approved by the Swedish ethics authority (reg. no 2021-06704-01) and is preregistered at Clinical-Trials.gov. The participants will receive written and verbal information about the study. The participants will digitally sign an informed consent in a secure platform.

### Participants

Participants will be recruited via advertisements in social media and with the help of relevant patient associations' mailing lists and websites. Interested individuals with self-reported AD diagnosis may participate following their written consent to the study. Advertisements in different types of social media has the advantage that it reaches different types of people regarding age and gender. In some platforms, it is possible to make the advertisements to only appear for men, if the female percentage should be considered too high, which is something we could consider. In order to minimise false diagnoses, participants will disclose at which point in time they received their AD diagnosis, and if they can recall, at what clinic and by which physician. They will then receive login information to access a secure digital platform, where data from self-assessment scales will be registered. Participation in the study does not involve any change in their regular treatment or medical self-care. Participants

will be evaluated by a licensed psychologist in a clinical interview, assessing suitability according to the inclusion criteria and motivation. If a potential participant states that they are interested, knowing that these interventions will require considerable time and effort, that participant is considered motivated.

### Inclusion criteria

At least 18 years of age with self-reported AD diagnosis. No new types of medications introduced for 6 months prior, with no intention of future change (participants will not be excluded if they change their medication after inclusion). Participants need to be able to understand written and spoken Swedish, have access to the internet and be able to receive telephone calls and text messages.

### Exclusion criteria

Other disease or condition that has immediate treatment priority over AD. This could be an ongoing demanding treatment of a severe somatic or psychiatric condition. No specific diagnoses are inevitably a cause for exclusion, as potential participants will rather be evaluated on a case-to-case basis.

## Setting

The trial is conducted at Karolinska Institutet, a medical university in Stockholm, Sweden. Dermatologists involved in the study are also associated with the dermatology clinic at Karolinska University Hospital, in Stockholm, Sweden. Initial assessment will be made by telephone by a clinical psychologist, and both study arms are entirely conducted online, including the collection of self-rated assessments. Participants will be recruited from all parts of Sweden.

## Power calculation

Based on data from Hedman-Lagerlöf *et al*[17] and Kern *et al*,[19] we expect an in-group effect of about d=0.85 for both interventions. A consultant statistician performed a power calculation. The required sample size was computed based on boot-strapped (5000 simulations) reruns of weekly measured data obtained in the therapist-guided RCT.[17] The results indicated that a sample size of 174 would be required for 90% power, α=0.05, given a non-inferiority margin of 3 points on the POEM scale and accounting for 10% missing data.

## Planned interventions

### Usual care

Participants will continue their usual care. According to national guidelines, people with AD should be treated according to a stepped-care model, with emollients and education at the bottom, advancing to cortisone treatments, up to more advanced treatments, such as immunosuppressive and biological interventions. Between individuals, usual care in Sweden can vary greatly. People with AD in Sweden may be treated and followed up at primary or specialist levels. Not all people with AD have regular contact with a physician at all, some have intermittent check-ups, and others may have close and frequent contact. Psychological interventions are not usually available, whereas educational interventions may be available in some regions of Sweden.

### Guided care

The aim of CBT for AD is to target behavioural and emotional processes, to improve AD symptoms and increase quality of life. The participants read education material, and use worksheets and homework reports. The material is in Swedish and is currently only available to study personnel and participants. This intervention consists of 10 modules and lasts 12 weeks. An introduction provides information about CBT and AD. Participant register itching and scratching behaviours, with the aim to increase awareness of these behaviours. Gradually, the participants are introduced to the proposed active treatment components: mindfulness training and exposure with response prevention. Mindfulness is trained throughout the intervention. The purpose of mindfulness is to increase awareness of emotional and behavioural reaction patterns, and to increase tolerance of aversive inner experiences.[21] Additionally, mindfulness training may improve stress tolerance. Participants are also trained in self-compassion, to increase understanding of one's own reactions and to increase the ability to handle difficult situations in the treatment and life.[22] After practising these fundamental skills, exposure with response prevention is introduced in week 4. Exposure is a controlled way to gradually approach situations that evoke strong emotions or somatic reactions. The purpose is to gain knowledge and experience in how this approach can change one's perception of unwanted situations, emotions or somatic reactions.[23] The assumption is that skills learnt from mindfulness practice and self-compassion training helps the patient to conduct exposure exercises. Response prevention means to reduce scratching in these problematic situations in order to reduce itch in the long term. Through this whole process, the participant is guided by a therapist. The therapist may be contacted through a secure internet platform at any time and responds within 48 hours. Participants read the education material, construct exercises to perform, describe their progress in homework assignments and receive feedback from their therapist. During the 12-week period, the participants work at their own pace, and contacts their therapist when necessary. The recommended pace, however, is one module per week, and the therapist contact participants who are inactive. All participants have access to the same treatment material but will construct exposure exercises that fit their specific situation, assisted by the therapist. The therapists will guide participants to make exercises as faithful to the treatment as possible. Participants will report back what they have done. Because therapist and participants never meet, there is no control of whether participants actually perform exercises as they are reported. The study coordinator will offer guidance to other therapists if needed.

### Digital self-care

As with guided care, the aim of CBT for AD is to target behavioural and emotional processes, to improve AD symptoms and increase quality of life. The material is similar to guided care but shortened. Likewise, participants, read information and perform exercises in their everyday life. As with guided care, the main treatment components are mindfulness and exposure with response prevention, with a specific focus on itch. Participants will be able to practice mindfulness with the help of instructions and examples. Because it may be challenging to construct relevant exposure exercises without the help of a therapist, participants will receive detailed instructions and guidance, and get to choose exposure exercises from a long list of examples. They will also be free to create their own relevant exposure situations within the intervention. Digital self-care consists of eight modules. The only active interventions, mindfulness and exposure are introduced weeks 1 and 2, respectively. From weeks 3 to 7, no new interventions are introduced and participants are advised to continue practising the interventions, but there is a new theme of education each week, to act as inspiration for exercises. The last 4 weeks, the participant

is merely advised to continue practice. There is no therapist, but the study coordinator monitors the process and may contact inactive participants. Participants will access the same digital platform as the guided condition. Participants work entirely at their own pace. Modules will be unlocked automatically if 1 week has passed, and all material in the previous module has been read. Participants will report if they have done their assignments, but there is no control if they do as they report.

### Differences between study conditions

The main differences between the conditions are that digital self-care (A) has contact with a clinician at only two points in time: preintervention and after 8 weeks plus the addition of automated messages, whereas guided care has continuous contact throughout the treatment via text messages on a secure platform and that (B) the treatment text is significantly shortened. Automated emails are sent every week to the non-guided group, consisting of a greeting and some encouraging words relevant to the week's content. Both groups receive automated text messages at the time of new assessments. Guided care will also have an evaluation from a clinician via telephone at two points in time. Furthermore, digital self-care uses a mobile optimised interface, and have improvements in design aimed to decrease the need of guidance.

### Therapists

Only the guided group will have therapist contact. The self-guided group will be managed by the study coordinator. The therapists are licensed psychologists specialised in CBT. All therapists have experience in working with somatic conditions and in a text-based format. All therapists will participate in an introductory workshop before treating in the study. The project coordinator will supervise the therapists regularly and on-demand.

### Randomisation/blinding

Participants will be blinded to condition. They will be generally informed that they will be randomised to 'either one of two types of self-help interventions based on CBT', not knowing any difference between the conditions. After baseline measurements, included participants will be allocated randomly (1:1) to therapist-guided or self-guided treatment in consecutive even-numbered blocks. Randomisation will be based on a true random number service and conducted by a person blinded to the participants, who is not otherwise involved with the study, using a true random number generator (http://www. random.org). It is impossible for the psychologist performing the inclusion assessment to anticipate participant allocation.

### Measurements

The outcome measures are based on those used previously by Hedman-Lagerlöf et al[17] and Kern et al,[19] for the evaluation of CBT interventions for AD. The battery of instruments in the current study overlaps with the recently published harmonising measures for eczema criteria, but have several additional measures, and does not include

some of the suggested measures.[24] We do include the suggested measurements for self-reported symptoms and quality of life but have no clinician-rated measurement of symptoms, which we acknowledge as a weakness.

### Primary outcome measure

POEM[20] measures the severity of symptoms of AD, which has a range of 0–28. The POEM has excellent psychometric properties and is widely used.[25] In this study, in order to be conservative, response is operationalised as a POEM score reduction 4 points or more, and the non-inferiority margin is operationalised as 3 points. This is the same as earlier studies of CBT for AD, based on changes from severe eczema. Another evaluation has found smaller changes, around 2.1–2.9 as 'likely to be beyond measurement error but unlikely to be clinically important'.[26] Because of this, as we are including participants with mild, moderate and severe eczema, we could possibly have chosen an even smaller margin.

### Secondary outcome measures

Demographical data will be collected on screening. Peak pruritus Numerical Rating Scale[27] is used to measure itching sensations; Perceived Stress Scale[28] is used to measure general experience of stress; Patient Health Questionnaire-9[29] is used to measure depressive symptoms; Dermatological Life Quality Index[30] is used to measure AD-specific quality of life; Insomnia Severity Index[31] is used to measure severity of sleep problems and possible insomnia; Client Satisfaction Questionnaire-8[32] is used to measure how satisfied participants are with the intervention. We will also ask participants to report if they experience adverse events associated with the treatment (AE). In addition to this, we will use the Credibility/Expectancy Scale[33] to measure credibility. At the end of the intervention, the participants will be asked to estimate the Time spent in the Exercises (ToE). Euro-QOL Five Dimensions (EQ-5D)[34] will be used to assess quality of life from a health economics perspective. Treatment Inventory of Costs in Psychiatric Patients (TIC-P)[35] will be used to assess healthcare costs. Participant will also rate their user experience of the interventions with the System Usability Scale.[36] Resource use is measured by the time in minutes the clinician spends on the different tasks associated with administering the intervention, doing interviews, and giving support, or Time spent on Treating. Therapists will register their time spent throughout the intervention, and not at a specific measurement point.

Outcomes at all scales will be compared with available published results regarding other internet-based interventions for each scale.

### Measurement points

Please refer to table 1. There are no planned interim analyses. There are no predefined criteria for trial termination.

**Table 1** Measurement points

| Screening | Pre | Weekly | Week2 | MID | POST | FU6 | FU12 |
|---|---|---|---|---|---|---|---|
| Demo | POEM | POEM | CES | POEM | POEM | POEM | POEM |
| POEM | NRS | NRS | | NRS | NRS | NRS | NRS |
| NRS | DLQI | DLQI | | DLQI | DLQI | DLQI | DLQI |
| DLQI | PHQ-9 | PHQ-9 | | PHQ-9 | PHQ-9 | PHQ-9 | PHQ-9 |
| PHQ-9 | PSS | PSS | | PSS | PSS | PSS | PSS |
| | ISI | ISI | | TIC-P | TIC-P | TIC-P | TIC-P |
| | TIC-P | | | EQ-5D | EQ-5D | EQ-5D | EQ-5D |
| | EQ-5D | | | ISI | ISI | ISI | ISI |
| | | | | | CSQ-8 | | |
| | | | | | AE | | |
| | | | | | ToE | | |
| | | | | | SUS | | |

AE, adverse events; CES, Credibility/Expectancy Scale; CSQ-8, Client Satisfaction Questionnaire-8; Demo, demographic data; DLQI, Dermatological Life Quality Index; EQ-5D, Euro QOL 5 Dimensions; ISI, Insomnia Severity Index; NRS, Numerical Rating Scale; PHQ-9, Patient Health Questionnaire 9 item; POEM, patient-oriented eczema measure; PSS, Perceived Stress Scale; SUS, System Usability Scale; TIC-P, Treatment Inventory of Costs in Psychiatric Patients; ToE, Time (spent) on Exercises.

## Safety considerations

Participants' regular healthcare is unaffected by either of the study conditions. Possible benefits for patients with increased ability to manage their disease are presumed to justify any disadvantages in the form of time consumption and temporary discomfort. Adverse treatment effects will be examined after the total twelve weeks of the intervention, and will be acted on if they are spontaneously reported before. If the disease worsens, the participants will be encouraged to seek help from their regular specialist clinic or primary healthcare centre. All communication and data storage take place on an encrypted and secure digital platform and is in accordance with the Karolinska Institutet's regulations and the EU General Data Protection Regulation. In case of any serious harm, the standard patient's insurance for Government financed healthcare will be applicable to every participant.

## Statistical analysis

### Non-inferiority margin

Earlier research[22] has defined clinically significant improvement on the main outcome measure to be 3.4 points, which is the basis for our non-inferiority margin of 3 points.

### Non-inferiority

Change over time will be modelled using mixed-effects linear regression with a random intercept and slope (time). The main outcome POEM will be analysed by the regression on time (baseline to end point), group (digital self-care vs guided care) and the interaction of time and group. We will consider digital care to be non-inferior to therapist guided if the upper bound of the one-sided 95% CI for the time × group interaction ($\alpha = 0.05$) is within the a priori non-inferiority margin of 3 points on the POEM.[36] In addition, we will investigate non-inferiority in a sensitivity analysis per-protocol framework with data

from patients who will have initiated module 4 or further in either of the study arms. As for interpretation of results, if the two-sided 95% CI does not contain zero, this will be viewed as indicative of superiority of the corresponding treatment. We will estimate Cohen's d by dividing the time × group coefficient by the pooled observed baseline SD. All primary analyses will be conducted intent to treat. The power calculations allow for 10% missing post-treatment data on the primary outcome at 90% power. If we have more than 10% missing data, we will use multiple imputation before conducting the primary analysis.

### Resource utilisation

At the primary endpoint, resource utilisation will be preliminary assessed by measuring therapist time for each treatment condition and compared with a typical salary for a clinical psychologist employed in Stockholm, Sweden.

### Cost-effectiveness

At 1-year follow-up, a cost-effectiveness analysis will be performed from a healthcare system perspective, based on recorded therapist time, as well as from a societal perspective. The incremental cost-effectiveness ratio will be estimated at post-treatment and at follow-up. We will also use the TIC-P to collect economic cost data,[34] together with the EQ-5D.[35] The costs will be assessed in Swedish krona and converted into US dollar using 2023 as the reference year. The costs of the self-guided and therapist-guided treatments will be modelled as a function of therapist time using the same healthcare index to determine tariffs for licensed psychologists. According to our hypothesis, there should be a significant difference in treatment costs but there should not be any significant difference in societal costs. We will calculate quality-adjusted life-years using the area under the curve approach,[37] use regression analyses to analyse costs[38] and

use non-parametric bootstrapping with 5000 iterations, where differences in costs and outcomes will be paired. The incremental cost-effectiveness ratio will be used as a cost-effectiveness estimate.[39]

## Mediation

We are interested in the mediating effect of time spent on mindfulness and exposure exercises on the outcome. Because we have a non-inferiority hypothesis, we will not perform traditional mediation analysis where the mediator is proposed to account for between-group differences. Instead, we will do within-group process analysis to investigate if treatment activity predicts simultaneous and/or subsequent change in POEM during treatment.[40] The exact analytical model will be determined based on the nature of data that we are able to collect on treatment activity. We will also explore potential moderated mediation, that is, if mediators differ between the two types of treatment. We also intend to analyse self-rated pruritus, perceived stress and depressed mood as potential mediators.

## Data

All data will be stored at a secure digital platform, to which only study personnel have access. Measurements from the participants will be submitted in the platform itself. Because all data are submitted and stored in the same digital platform, we will not use a data monitoring committee. The final trial data set will be primarily handled by the first, second and final author. However, all authors will be granted access to the data set if necessary. We have no plans to make an individual-level data set or any statistical code public. Data will be available on reasonable request to the corresponding author.

## Patient and public involvement

We have used input from participants in the feasibility trial,[19] on which we based intervention improvements in the present study. This included various revisions, the most important of which was to add a 'scratch-free time' intervention. The duration is defined by the participant and gradually increased. This ensured an easier start compared with the previous variant to begin with an entire scratch-free day. Some participants also wanted therapist guidance, which could not be considered for this particular intervention. We also considered feedback from a patient representative, who was not a trial participant. They largely agreed with the feedback from trial participants, encouraged the intervention in its current form, and furthermore participated in an introductory seminar with the treating study psychologists. In the current trial, no participants were involved in decisions about study design, outcome measures or research question. A patient organisation for AD, as well as one for disorders related to asthma and allergies (Atopikerna and Astma & Allergiförbundet) will be involved in the recruitment of participants by informing their members about the study. We plan to disseminate the results of the research to study participants and to the aforementioned patient organisations.

## ETHICS AND DISSEMINATION

The study will be carried out according to the predefined standards of this trial protocol, the Declaration of Helsinki and Good Clinical Practice. This study will be reported according to the CONSORT statement for non-pharmacological trials.[41] The interventions that will be administered have deemed safe by consultant dermatologists and have been previously evaluated without serious AEs.[18 19] This study will use the Vancouver criteria for authorship. The results of this study will be published in peer-reviewed scientific journals, be presented to the scientific community at conferences. The results will also be communicated to study participants, patient organisations and clinicians.

**Contributors** DK: study design, intervention design, drafting of manuscript, planning of statistical analysis. BL: study design, intervention design, planning of statistical analysis. LL: contributions to intervention and manuscript; MB: study concept and contributions to manuscript. EH-L: study concept, contributions to manuscript and planning of statistical analysis; NL: study concept, study design, contributions to manuscript. MK: study concept, contributions to manuscript, intervention design, study design, planning of statistical analysis, principal investigator. All authors: critical review and revision of manuscript.

**Funding** This study will be funded in part by the Swedish Ministry of Health and Social affairs.

**Disclaimer** The funder has no role in the design and conduct of the study and will have no influence of data collection, management, analysis, interpretation. The funder will have no part in preparation, review or approval of the manuscript, or the decision to submit the manuscript for publication.

**Competing interests** BL reported being a shareholder of DahliaQomit AB and holding copyright for a cognitive behavioural treatment manual for irritable bowel syndrome, with royalties paid from Pear Therapeutic, outside of submitted work. LL reported receiving a personal fee from LEO Pharma and Sanofi Genzyme outside of submitted work. EH-L reported being a shareholder of DahliaQomit AB and holding copyright for a cognitive behavioural treatment manual for irritable bowel syndrome, with royalties paid from Pear Therapeutic, Inc, outside of submitted work. No other disclosures were reported.

**Patient and public involvement** Patients and/or the public were involved in the design, or conduct, or reporting, or dissemination plans of this research. Refer to the Methods section for further details.

**Patient consent for publication** Not applicable.

**Provenance and peer review** Not commissioned; externally peer reviewed.

**ORCID iD**
Dorian Kern http://orcid.org/0000-0003-2012-8186

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
