## [Reviewer comments · BMJ Open]

ARTICLE DETAILS

TITLE (PROVISIONAL)	Brief Self-guided Digital Intervention versus a Comprehensive Therapist-guided Online Cognitive Behavioral Therapy for Atopic Dermatitis: A Trial protocol for a Randomized Non-Inferiority Trial
AUTHORS	Kern, Dorian; Ljótsson, Brjánn; Lönndahl, Louise; Hedman-Lagerlöf, Erik; Bradley, M; Lindefors, Nils; Kraepelien, Martin

VERSION 1 – REVIEW

REVIEWER	Krieger, Tobias Universität Bern Philosophisch-humanwissenschaftliche Fakultät
REVIEW RETURNED	24-Oct-2022

GENERAL COMMENTS	The authors present a study protocol of a non-inferiority randomized controlled trial comparing a guided ICBT and an unguided shortened form of this program. The manuscript is timely and generally well-written. However, I have some comments on the present form of the manuscript that the authors might want to consider. - Title: It might be clearer what is compared in the study if the title included the term „unguided“ or „self-guided“.- P5: „One solution is to use existing ICBT programs and simply remove the therapist-guidance, but previous research has suggested that this has negative impact on effectiveness“ -> I am not sure whether this conclusion can be drawn from the referenced study since it is on the comparison of guided self-help and face-to-face therapy I think that the authors could find a more suitable reference for their statement.- P6: For reasons of consistency, „non-inferiority“ not „noninferiority“- P7: „Participants will be evaluated by a licensed psychologist in a clinical interview, assessing suitability and motivation.“ ==> Does this mean that a lack of motivation would lead to the exclusion of a patient?- P7: Who will conduct the assessments? Dermatologists or clinical psychologists? Which assessments?- P9: Automated messages are mentioned in the self-care condition. Could the authors elaborate a bit more on when these messages are sent and on the content of these messages?P11: Not all measures that are listed in below are listed here: NPR vs. NRS. This may be a mistake. In addition, sometimes measures are described in a full sentence, sometimes not. I would ask the authors for more consistency.- P11: The information on which measures are assessed at which measurement points could be presented in a table.- P13: "Significant missing data will be analyzed using multiple imputation." In my opinion this sentence does not make much
---

	sense. Does it mean that before the authors conduct the primary analysis, they impute missing data? Please describe in more detail what you mean. Also, a statement that primary analyses will be performed on the intention-to-treat analysis would help. - P14: I am not sure what the asterisk in line 5 stands for. Discussion: Since the authors manipulated two aspects of the original ICBT program and its delivery, respectively, i.e., content and guidance, in case of non non-inferiority, it will not be clear which of these factors have led to inferiority. This could be discussed at the end of the manuscript.
--	--

REVIEWER	Santer, Miriam University of Southampton, Primary Care & Population Sciences
REVIEW RETURNED	09-Dec-2022

GENERAL COMMENTS	Thank you very much for inviting me to review this protocol for an important RCT. It is clearly written and well set out. I feel this protocol paper would be strengthened if the authors could address the following points. In 'strengths and limitations of this study' the authors acknowledge the risk of bias towards recruitment of female participants with an advanced education. It would therefore be good to see discussion of how this will be addressed in their Methods section. There is little mention of existing eczema literature. There are at least 2 relevant systematic reviews as well as at least one recent online intervention for eczema (see below). Although the focus of other online interventions differ, it would still be appropriate to acknowledge. Inclusion criteria states, "No new types of medications introduced for six months prior, with no intention of future change." This requires further explanation. Some future change is to be expected? What would happen if there was a change? It would be helpful to describe the interventions in line with TIDIER guidance. Although I understand from the abstract that both groups continue to receive usual care, it would be useful to reiterate this in the 'planned interventions' section and to describe what usual care consists of in Sweden, as provision of eczema services varies substantially between countries, including availability of psychological and educational support. Patient and public involvement section needs greater clarity on the distinction between public involvement in research vs feedback from research participants, as these are two separate things. The authors mention that a patient representative offered feedback on the feasibility trial but it would be useful to outline specifically what changes were made, in the spirit of GRIPP-2 reporting. The authors write that, "The outcome measures are based on those used previously by Hedman-Lagerlöf and colleagues[17] and Kern and colleagues[18], for the evaluation of CBT interventions for AD." It would be appropriate to refer to the international core outcome set for eczema trials (Harmonising Outcome Measures for Eczema) and justify why their chosen outcomes do or don't align with the core outcome set. The paper they refer to by Schram and colleagues is based on severe eczema and more recent MCID research on the POEM suggests
--

	smaller differences may still be important (see Howells et al below). A limitation of this non-inferiority study is that they are including mild, moderate and severe eczema but have based their non-inferiority margin on severe eczema. References Howells L, Ratib S, Chalmers JR, Bradshaw L, Thomas KS, CLOTHES trial team. How should minimally important change scores for the Patient-Oriented Eczema Measure be interpreted? A validation using varied methods. British Journal of Dermatology. 2018 May;178(5):1135-42. Pickett K, Loveman E, Kalita N, Frampton G, Jones J. Educational interventions to improve quality of life in people with chronic inflammatory skin diseases: systematic reviews of clinical effectiveness and cost-effectiveness. Health Technology Assessment. 2015;19(86):i-176. Ersser SJ, Cowdell F, Latter S, Gardiner E, Flohr C, Thompson AR, Jackson K, Farasat H, Ware F, Drury A. Psychological and educational interventions for atopic eczema in children. Cochrane Database of Systematic Reviews. 2014(1). Santer M, Muller I, Becque T, Stuart B, Hooper J, Steele M et al. Eczema Care Online behavioural interventions to support self-care for children and young people: two independent, pragmatic, randomised controlled trials BMJ 2022; 379 :e072007 doi:10.1136/bmj-2022-072007
--	--

VERSION 1 – AUTHOR RESPONSE

Reviewer: 1

Dr. Tobias Krieger, Universitat Bern Philosophisch-humanwissenschaftliche Fakultat

Comments to the Author:

The authors present a study protocol of a non-inferiority randomized controlled trial comparing a guided ICBT and an unguided shortened form of this program. The manuscript is timely and generally well-written. However, I have some comments on the present form of the manuscript that the authors might want to consider.

- Title: It might be clearer what is compared in the study if the title included the term „unguided“ or „self-guided“.

Response: As per reviewer suggestion, the title is changed to: “Brief Self-guided Digital Intervention versus a Comprehensive Therapist-guided Online Cognitive Behavioral Therapy for Atopic Dermatitis: A Trial protocol for a Randomized Non-Inferiority Trial”

- P5: „One solution is to use existing ICBT programs and simply remove the therapist-guidance, but previous research has suggested that this has negative impact on effectiveness“ -> I am not sure whether this conclusion can be drawn from the referenced study since it is on the comparison of guided self-help and face-to-face therapy I think that the authors could find a more suitable reference for their statement.

Response: We agree with the reviewer and thank him for pointing this out. We reconstructed the sentence and used the next reference for this statement: “One solution is to use existing ICBT

programs and simply remove the therapist-guidance. Some research has suggested that this has negative impact on effectiveness, however, there are indications that the amount and type of therapist contact are not decisive[14] and recent studies have compared therapist supported ICBT and self-guided ICBT and shown that treatment outcome, treatment implementation and treatment satisfaction do not differ between conditions if certain components are present in the self-guided intervention.

- P6: For reasons of consistency, „non-inferiority“ not „noninferiority“

Response: A hyphen has been added.

- P7: „Participants will be evaluated by a licensed psychologist in a clinical interview, assessing suitability and motivation.“ ==> Does this mean that a lack of motivation would lead to the exclusion of a patient

Response: We agree that this description was unclear. We don't make a formalized evaluation of motivation, nor do we exclude anyone who states their interest in participating who otherwise fulfill the criteria. We added : "At the end of the interview, the interviewer explains that participation will require considerable effort, and asks if the potential participants believes this to be reasonable. If the potential participant answers yes, the interviewer formally asks if they are interested study participation. If a potential participant states that they are interested, acknowledging that these interventions will require considerable time and effort, that participant is considered motivated."

- P7: Who will conduct the assessments? Dermatologists or clinical psychologists? Which assessments?

Response: Initial assessments will be conducted by a licensed psychologist. A statement was added to that effect.

- P9: Automated messages are mentioned in the self-care condition. Could the authors elaborate a bit more on when these messages are sent and on the content of these messages?

Response: The following was added: "Automated emails are sent every week to the non-guided group. These email consists of a greeting from the research team, and a few paragraphs of guidance regarding the week's treatment content. These emails are an effort to maintain a sense of contact with the research team throughout the intervention. Both groups receive automated reminder text messages at the time of new assessments."

P11: Not all measures that are listed in below are listed here: NPR vs. NRS. This may be a mistake. In addition, sometimes measures are described in a full sentence, sometimes not. I would ask the authors for more consistency.

Response: We thank the reviewer for pointing this out. The correct abbreviation is NRS. This has been changed throughout. All measures are now described in a full sentence.

- P11: The information on which measures are assessed at which measurement points could be presented in a table.

Response: We thank the reviewer for his suggestion. A table has been added.

- P13: "Significant missing data will be analyzed using multiple imputation." In my opinion this

sentence does not make much sense. Does it mean that before the authors conduct the primary analysis, they impute missing data? Please describe in more detail what you mean. Also, a statement that primary analyses will be performed on the intention-to-treat analysis would help.

Response: We agree that statement wasn't clear. We have explained this further at the end of the "Non-inferiority" section: "All primary analyses will be conducted intent-to-treat. The power calculations allow for 10% missing post-treatment data on the primary outcome at 90% power. If we have more than 10% missing data, we will use multiple imputation before conducting the primary analysis."

- P14: I am not sure what the asterisk in line 5 stands for.

Response: This was an error and has been removed.

Discussion: Since the authors manipulated two aspects of the original ICBT program and its delivery, respectively, i.e., content and guidance, in case of ~~non~~ non-inferiority, it will not be clear which of these factors have led to inferiority. This could be discussed at the end of the manuscript.

Response: We thank the reviewer for his insightful comment. A comment about has been added to the strengths and limitations section. "The self-guided condition is different from the therapist-guided in several ways, including no therapist and shorter content. If the self-guided condition cannot be found non-inferior, this study cannot differentiate what factors caused that result."

Reviewer: 2

Dr. Miriam Santer, University of Southampton

Comments to the Author:

Thank you very much for inviting me to review this protocol for an important RCT. It is clearly written and well set out. I feel this protocol paper would be strengthened if the authors could address the following points.

In strengths and limitations of this study the authors acknowledge the risk of bias towards recruitment of female participants with an advanced education. It would therefore be good to see discussion of how this will be addressed in their Methods section.

Response: We have added this part to the "participants section": "Advertisements in different types of social media has the advantage that it reaches different types of people regarding age and gender. In some platforms, it is possible to make the advertisements to only appear for men, if the female percentage should be considered too high, which is something we could consider."

There is little mention of existing eczema literature. There are at least 2 relevant systematic reviews as well as at least one recent online intervention for eczema (see below). Although the focus of other online interventions differ, it would still be appropriate to acknowledge.

Response: We are very grateful for these articles. We have acknowledged them in the introduction section, under the new heading "Educational and psychological interventions for atopic dermatitis. In this instance, we chose to omit Ersser et al, however, as we only focus on adults in this study: "Various forms of educational and psychological support may be available for people with AD. According to structured reviews, studies of educational interventions for AD have often had suboptimal methodology and unclear results, with tendencies towards effect on skin-related quality of life and improved symptoms[8]. These interventions may be held in a group or individual setting and

may be led by physicians, nurses or other professionals. One recent RCT of an Internet-delivered educational and behavioral intervention, including young adults, showed small but significant effects on AD symptoms and secondary measures [9].”

Inclusion criteria states, “No new types of medications introduced for six months prior, with no intention of future change.” This requires further explanation. Some future change is to be expected? What would happen if there was a change?

Response: If a participant were about to start a new medical intervention, this would interfere with the results of the investigated interventions. We would like to minimize this by not including any participants who are about to change medication or try another intervention, such as phototherapy. After intervention start, however, we do not control for this, and no study participation will be terminated because of new medications. However, at post-treatment, we ask patient about potential change in medications during the treatment period and this will be reported.. This is explained in the inclusion criteria: ” No new types of medications introduced for six months prior, with no intention of future change (participants will not be excluded if they change their medication after inclusion).”

It would be helpful to describe the interventions in line with TIDIER guidance. Although I understand from the abstract that both groups continue to receive usual care, it would be useful to reiterate this in the ‘planned interventions’ section and to describe what usual care consists of in Sweden, as provision of eczema services varies substantially between countries, including availability of psychological and educational support.

Response: We thank the reviewer for her important comments. We have expanded and restructured the intervention descriptions with the TIDIER guidelines in mind. Usual care in Sweden can vary greatly within Sweden and between individuals. Usual care essentially means whatever care they would otherwise receive. This has been elaborated on in “planned interventions”.

Patient and public involvement section needs greater clarity on the distinction between public involvement in research vs feedback from research participants, as these are two separate things. The authors mention that a patient representative offered feedback on the feasibility trial but it would be useful to outline specifically what changes were made, in the spirit of GRIPP-2 reporting.

Response: We thank the reviewer for her feedback. We have made an effort to make that section more clear, with added information, for example: “We also considered feedback from a patient representative, who was not a trial participant. They largely agreed with the feedback from trial participants, encouraged the intervention in its current form, and furthermore participated in an introductory seminar with the treating study psychologists.”

The authors write that, “The outcome measures are based on those used previously by Hedman-Lagerlöf and colleagues[17] and Kern and colleagues[18], for the evaluation of CBT interventions for AD.” It would be appropriate to refer to the international core outcome set for eczema trials (Harmonising Outcome Measures for Eczema) and justify why their chosen outcomes do or don’t align with the core outcome set. The paper they refer to by Schram and colleagues is based on severe eczema and more recent MCID research on the POEM suggests smaller differences may still be important (see Howells et al below). A limitation of this non-inferiority study is that they are including mild, moderate and severe eczema but have based their non-inferiority margin on severe eczema.

Response: We are most grateful that the reviewer shared her insight and these articles with us. According to reviewer suggestion, we discuss the HOME criteria and the implications of the Howells

article under the Measurements heading. Under the Primary outcome measure heading, we also discuss our non-inferiority margin: “This is the same as earlier studies of CBT for AD, based on changes from severe eczema. Another evaluation has found smaller changes, around 2.1 to 2.9 as “likely to be beyond measurement error but unlikely to be clinically important”[26]. Because of this, as we are including participants with mild, moderate and severe eczema, we could possibly have chosen an even smaller margin.”

References

Howells L, Ratib S, Chalmers JR, Bradshaw L, Thomas KS, CLOTHES trial team. How should minimally important change scores for the Patient-Oriented Eczema Measure be interpreted? A validation using varied methods. British Journal of Dermatology. 2018 May;178(5):1135-42.

Pickett K, Loveman E, Kalita N, Frampton G, Jones J. Educational interventions to improve quality of life in people with chronic inflammatory skin diseases: systematic reviews of clinical effectiveness and cost-effectiveness. Health Technology Assessment. 2015;19(86):i-176.

Ersser SJ, Cowdell F, Latter S, Gardiner E, Flohr C, Thompson AR, Jackson K, Farasat H, Ware F, Drury A. Psychological and educational interventions for atopic eczema in children. Cochrane Database of Systematic Reviews. 2014(1).

Santer M, Muller I, Becque T, Stuart B, Hooper J, Steele M et al. Eczema Care Online behavioural interventions to support self-care for children and young people: two independent, pragmatic, randomised controlled trials BMJ 2022; 379 :e072007 doi:10.1136/bmj-2022-072007

Hywel C Williams, Jochen Schmitt, Kim S Thomas, Phyllis Spuls, Eric L Simpson, Christian J Apfelbacher, Joanne R Chalmers, Masutaka Furue, Norito Katoh, Louise A Gerbens, Yael A Leshem, Laura Howells, Jasvinder A Singh, Maarten Boars, HOME initiative J Allergy Clin Immunol. 2022 Jun;149(6):1899-1911. doi: 10.1016/j.jaci.2022.03.017 Epub 2022 Mar 26.

VERSION 2 – REVIEW

REVIEWER	Santer, Miriam University of Southampton, Primary Care & Population Sciences
REVIEW RETURNED	06-Feb-2023
GENERAL COMMENTS	Thank you for addressing these comments so comprehensively